# Rethinking the Definition of Unlearning: Suppressive Machine Unlearning

## Abstract

Machine unlearning, an emerging issue of privacy concern in the deep learning era, is practically motivated by the *data removal* from training or *knowledge suppression* of utility on that data. Unfortunately, retraining via data removal, which has been understood as the gold standard, does not elucidate how much we suppress the model's knowledge on the target. The existing definition well covers an *exact* or *approximate* unlearning only with a removal perspective, yet failing to encompass knowledge suppression incurred via unlearning. Moreover, suppression is tightly entangled with removal in a way that more knowledge suppression obviously leads to significant divergence from exact and approximate unlearning, thus motivating us to rethink the definition of machine unlearning. We formally introduce a novel definition of *Suppressive Machine Unlearning*, encompassing how far the unlearned model is from retraining, i.e., $(\varepsilon, \delta)$-approximate unlearning, and how much the model's utility becomes suppressed, i.e., $\kappa$. To illuminate the formal dynamics between removal and suppression, we reveal the trade-off between the removal guarantees $(\varepsilon, \delta)$, which quantifies how much it deviates from an idealized retraining and $\kappa^*$, which is the requested level of suppression.

## 1 Introduction

Modern machine learning systems are trained on vast, heterogeneous corpora—spanning text, images, code, and audio—that blend proprietary datasets with data scraped from the open web. As these systems become pervasive across consumers and enterprises, they encounter diverse governance demands (Cloud Security Alliance, AI Governance & Compliance Working Group, 2024; European Data Protection Supervisor, 2025): organizations face requests to remove the influence of certain examples, individuals seek to retract personal data, and model owners aim to align content with evolving policies or prevent the exposure of certain capabilities in high-risk contexts (Bai et al., 2022; Yao et al., 2024). In short, *what should be removed* and *what should be suppressed* are now first-class operational questions, no longer afterthoughts. *Machine unlearning* (Cao & Yang, 2015; Liu et al., 2025) has emerged as a promising solution to address these demands, aiming to modify a trained model so that it behaves as if certain examples had never been included during training.

These diverse unlearning demands fundamentally align with two distinct, yet often interconnected, objectives: *Data Removal* and *Knowledge Suppression*. Requirements stemming from privacy and consent concerns (e.g., the "right-to-be-forgotten" (Dang, 2021)) clearly necessitate data removal, expunging specific user-contributed examples that were validly collected but later withdrawn. On the other hand, evolving intellectual property and licensing agreements, alongside dynamic product and platform policies, introduce category-level restrictions that cannot be straightforwardly reduced to merely removing a "list of training records" (Jia et al., 2021). This scenario primarily demands knowledge suppression, where the model is required to diminish specific capabilities, biases, or information related to certain concepts or content categories, even if the individual data remains.

From an engineering standpoint, these demands surface under tight constraints. Full retraining on a "retain-only" dataset is often computationally infeasible for large models and incompatible with rapid release cadences. Even when retraining is possible, the goal behind many requests is not simply to match a retrained parameter distribution. **More fundamentally, there is no agreed-upon definition of what and how constitutes such unlearning requests, especially when distinguishing between data removal and knowledge suppression.**

This complexity is compounded by stochastic optimization: differences induced by data removal can be masked by run-to-run variability, batch order, and optimizer noise, making the unlearned model hard to distinguish from the original on distributional tests that look only at parameters (Thudi et al., 2022b). This tension has motivated rigorous notions of unlearning that target *retraining equivalence*—exactly or approximately matching the distribution of model parameters produced by training on the retain-only dataset (Guo et al., 2019; Bourtoule et al., 2021; Nguyen et al., 2025). While such definitions provide crisp, auditable guarantees, they can not directly relate the broader intent of knowledge suppression, i.e., ensuring a model will stop answering certain questions or exposing specific internal features. In parallel, the field has seen the emergence of two complementary research lines: one focused on erasing internal knowledge (Thudi et al., 2022a; Jang et al., 2022; Gandikota et al., 2023; Fan et al., 2023), and the other on suppressing exposure on forget data while preserving capability elsewhere (Chen et al., 2023; Li et al., 2024b; Takashiro et al., 2024).

This state of affairs motivates a conceptual and formal re-examination of what it means to "unlearn," specifically through the lens of data removal and knowledge suppression. In a nutshell, we argue that unlearning mechanisms should satisfy two qualitatively distinct requirements, either individually or simultaneously: (i) *Data Removal* (the erasure of a forget set from the model), and (ii) *Knowledge Suppression* (a reduction in the model's utility on a forget-target distribution relative to an unrelated reference). In this paper, we propose a novel definition of machine unlearning–Suppressive Machine Unlearning, a unified theoretical framework that reconciles the tension between data removal and knowledge suppression in machine unlearning. We make the following key contributions:

- **Conceptual Framework for Unlearning Demands.** We formalize the distinction between Data Removal and Knowledge Suppression, and propose a taxonomy of three types of unlearning requests that capture the spectrum of real-world demands.

- **Unified Definition of** $(\varepsilon, \delta, \kappa)$**-Suppressive Machine Unlearning.** We introduce a novel definition that simultaneously guarantees $(\varepsilon, \delta)$-approximate data removal at the mechanism level and $\kappa$-suppression at the operational level. This bridges the gap between deletion/suppression-centric approaches under a single framework.

- **Characterization of the Removal-Suppression Relationship.** Through both theoretical analysis and empirical validation, we prove that data removal parameters $(\varepsilon, \delta)$ directly bound the achievable suppression level $\kappa$, and empirically validate these theoretical predictions across multiple unlearning methods.

## 2 PRELIMINARIES

Early work to define machine unlearning intuitively focused on the outcome that "an unlearned model should be indistinguishable from a model retrained without the forget data." However, this notion soon revealed several limitations: (i) indistinguishability is ambiguous at the single-model level, (ii) it was overly permissive in practice, and (iii) it proved practically unverifiable. Thus, recent studies have evolved into a mechanism-perspective definition of unlearning that deals with probabilistic guarantees of unlearning mechanisms, rather than the resulting model itself.

Let $\mathcal{X}$ be the input space and $\mathcal{X}^*$ be the set of all possible training datasets. For a dataset $D \in \mathcal{X}^*$ and a forget set $D_f \subseteq D$. Let $\mathcal{H}$ be the model hypothesis space. A learning algorithm is a mapping $A : \mathcal{X}^* \rightarrow \mathcal{H}$. An unlearning mechanism is $U$ that, given $(D, D_f, A(D))$, outputs a (possibly randomized) model in $\mathcal{H}$. For a random model $M$, we denote $\Pr(M \in T)$ as the probability that $M$ falls in a measurable set $T \subseteq \mathcal{H}$. Based on this setup, we recall the definition of exact unlearning:

**Definition 2.1** (Exact Unlearning (Nguyen et al., 2025)). *Given a learning algorithm $A$, a dataset $D$, and a forget set $D_f \subseteq D$, the unlearning $U$ achieves* exact unlearning *if and only if (iff)*

$$\forall T \subseteq \mathcal{H} : \quad \Pr\big(A(D \setminus D_f) \in T\big) \ = \ \Pr\big(U(D, D_f, A(D)) \in T\big). \tag{1}$$

*Equivalently, the output, i.e., unlearned model, distribution of $U(D, D_f, A(D))$ is identical to that of retraining model via $A$ on $D \setminus D_f$.*

**Remark.** The indistinguishability condition of exact unlearning can be applied either in the model's parameter space or, alternatively, in the output space. When applied to the output space, it requires the distributions of model outputs to be identical.

Intuitively, exact unlearning guarantees that the unlearned model behaves as if the forget data had never been used for model training, thereby eliminating any residual effect of the removed samples. However, achieving such a strong guarantee is often infeasible in practice due to extreme computational, storage, and auditability constraints (Xu et al., 2024). This practical challenge has motivated the development of approximate unlearning, which relaxes the strict equality requirement of exact unlearning by providing guarantees for the probabilistic indistinguishability (Guo et al., 2019). This notion is inspired by the framework of differential privacy (DP) (Dwork et al., 2006; 2014).

DP (Dwork et al., 2006) provides formal guarantees that the outputs of randomized mechanisms are probabilistically indistinguishable whether or not a single individual's data is included in the input datasets. By adapting this principle, Guo et al. (2019) introduced $(\varepsilon, \delta)$-*certified removal*, which formalizes *approximate unlearning* as a tractable relaxation of exact unlearning by bounding the statistical influence of a forgotten sample on the resulting model distribution.

**Definition 2.2** ($(\varepsilon, \delta)$-Approximate Unlearning (Neel et al., 2021))**.** *For $\varepsilon \geq 0$ and $\delta \in [0, 1)$, $U$ performs $(\varepsilon, \delta)$-certified removal for a learning algorithm $A$ if for all measurable $T \subseteq \mathcal{H}$, all datasets $D \in \mathcal{X}^*$, and any sample $z \in D$, we satsify:*

$$\Pr\big(U(D, z, A(D)) \in T\big) \; \leq \; e^{\varepsilon} \Pr\big(A(D \setminus \{z\}) \in T\big) + \delta,$$

*and*

$$\Pr\big(A(D \setminus \{z\}) \in T\big) \; \leq \; e^{\varepsilon} \Pr\big(U(D, z, A(D)) \in T\big) + \delta. \tag{2}$$

Both exact and approximate unlearning are typically defined over the parameter space, but weaker notions have been proposed in the output space (Baumhauer et al., 2022), where the objective is to bound the influence of the forget data on the model's predictions rather than its parameters. However, when unlearning is defined only in the output space, the guarantee can become indistinguishable from mere obfuscation (Hu et al., 2024); it is not empirically distinguishable whether the forget information is truly erased or simply masked with obfuscated outputs at decision time. To address this limitation, we separate the unlearning requirements by formalizing **Data Removal** and **Knowledge Suppression** as distinct conditions.

## 3  RETHINKING UNLEARNING: FROM DATA REMOVAL TO KNOWLEDGE SUPPRESSION

Consider the following unlearning request: *Alice (Data Owner) asks Bob (Model Owner) to "Erase my data (or knowledge about me) from the model."* The canonical goal of machine unlearning is for the unlearned model to be statistically indistinguishable from a model retrained from scratch without Alice's data. We formalize this as the $(\varepsilon, \delta)$-**Data Removal Condition**, which requires an unlearning mechanism $U$ to satisfy the definition of $(\varepsilon, \delta)$-Approximate Unlearning (Eq. 2).

However, this formulation, while crucial for ensuring the removal of data's influence, faces several frictions in practice. First, the formulation of $(\varepsilon, \delta)$-Approximate Unlearning is not sufficient for **Knowledge Suppression Condition**, where knowledge in a model typically arises as a generalized pattern or an emergent capability learned from a distribution of training data, not a property of a single data point. A model can satisfy the Data Removal Condition by eliminating the statistical influence of a specific input, yet still retain the broader knowledge that Alice wants suppressed. Second, in modern large-scale (pre-training) regimes, full retraining is often operationally infeasible, and even retrained models can preserve broad generalizations that still enable inference about the forget target. For example, Thudi et al. (2022b) formalize the notion of *data forgeability*, showing how minibatch SGD and data-order variability can render the unlearned model indistinguishable from the original, diluting the practical value of the guarantee. Thus, some recent approaches (Ji et al., 2024; Li et al., 2024a; Zhang et al., 2025; Li et al., 2025) focus on the practical behavior change rather than data-centric guarantees by enforcing refusals or degrading responses to target knowledge. These challenges highlight the need for an additional formulation that captures the unlearned model at the level of **Knowledge Suppression Condition**. We define this condition directly on the properties of a single (post-unlearning). We will establish its connection to unlearning mechanisms in the following section.

**Definition 3.1** (Knowledge Suppression Condition (single-model))**.** *Let $\mathcal{X}$ be the input space and $\mathcal{Y}$ the response space. Given two distributions over inputs: (i) a* forget-target *distribution $\mathbb{Q}_f$, and (ii)*

a reference *distribution* $\mathbb{Q}_0$. *Let $P_{\theta_u}(\cdot \mid x)$ denote the output distribution of an unlearned model $\theta_u$ on $\mathcal{Y}$ for a given $x$, and $s : \mathcal{X} \times \mathcal{Y} \to [0, 1]$ be a score function. A model $\theta_u$ satisfies $\kappa$-suppression for a level $\kappa \geq 0$ if:*

$$\mathbb{E}_{x_0 \sim \mathbb{Q}_0} \mathbb{E}_{y_0 \sim P_{\theta_u}(\cdot | x_0)} [s(x_0, y_0)] \; - \; \mathbb{E}_{x_f \sim \mathbb{Q}_f} \mathbb{E}_{y_f \sim P_{\theta_u}(\cdot | x_f)} [s(x_f, y_f)] \; \geq \; \kappa. \tag{3}$$

The suppression level $\kappa$ quantifies how much worse the model must perform on the forget-target distribution $\mathbb{Q}_f$ relative to the reference distribution $\mathbb{Q}_0$. A larger $\kappa$ enforces a stronger suppression of the targeted knowledge (greater performance drop on $\mathbb{Q}_f$), while $\kappa = 0$ corresponds to no guaranteed suppression beyond parity with $\mathbb{Q}_0$. The definition of $\kappa$-suppression for an unlearned model $\theta_u$ ensures that suppression is targeted and prevents the trivial solution of indiscriminately degrading the model's overall performance. For example, in a face-classification task with label space $Y = \{1, \ldots, C\}$, if the goal is to suppress knowledge of Alice's face, then $\mathbb{Q}_f$ would consist of her images, and $\kappa$-suppression would require the model's top-1 accuracy as a score function $s$ on $\mathbb{Q}_f$ to be at least $\kappa$ lower than its accuracy on general images from $\mathbb{Q}_0$. for LLM question answering, $\mathbb{Q}_f$ could contain prompts on a harmful topic, with $\kappa$-suppression requiring performance for a score function $s$ (e.g., ASR/ROUGE) on $\mathbb{Q}_f$ to be at least $\kappa$ lower than on benign QA prompts from $\mathbb{Q}_0$.

**Remark.** The response space $\mathcal{Y}$ is intentionally abstract: it may represent final outputs (e.g., tokens, labels, actions) or intermediate features (e.g., embeddings, logits). This allows suppression to be formalized either at the output level or in internal representations, enabling the notion to generalize across tasks and model architectures. The score function $s$ is initiated to match the operational target, such as accuracy or refusal probability for classification, precision or a token-level log-likelihood for generation, or a distance between hidden embeddings for representation-level. Since $s \in [0, 1]$, any bounded monotone transform is admissible.

Leveraging two conditions for unlearning, **Data Removal** and **Knowledge Suppression**, we can categorize unlearning requests into three types:

1. **Type I Request:** "Just erase it." This is exactly what the **Data Removal Condition** formalizes: the probability distribution of the unlearned model must be (exactly or approximately) indistinguishable from the probability distribution of the model retraining on $D \setminus D_f$.

2. **Type II Request:** Sometimes Alice demands more: the model must suppress knowledge about Alice's data beyond a given threshold $\kappa$ (as evaluated by the criterion $s$), and simultaneously guarantee that it has forgotten that data. This, in turn, requires satisfying both the **Data Removal Condition** and **the Knowledge Suppression** Condition.

3. **Type III Request:** Unlike Types I–II, no Data Removal Condition is claimed here. The goal is purely operational: enforce the **Knowledge Suppression Condition** at a $\kappa^*$ for a policy-defined target, while preserving general capability on non-target inputs. This setting also covers cases where Alice does not provide a concrete forget set $D_f$ (i.e. Zero-shot Machine Unlearning (Chundawat et al., 2023)).

## 4 SUPPRESSIVE MACHINE UNLEARNING

The definitions of machine unlearning (Eq. 1, 2) provide a rigorous, mechanism-level formulation of the Data Removal Condition. However, the growing body of work on practical suppression and refusal techniques cannot be fully subsumed by this formulation. The field currently lacks a single, comprehensive definition capable of representing the diverse unlearning requests we outlined in Section 3. In this section, we bridge this gap by unifying these two fundamental goals of unlearning-the **Data Removal** and **Knowledge Suppression** conditions-into a definition from a mechanistic perspective: **Suppressive Machine Unlearning**.

**Definition 4.1** (($\varepsilon, \delta, \kappa$)-Suppressive Machine Unlearning). *An unlearning mechanism $U$ is a randomized algorithm that maps $(D, z, A(D))$ to a distribution over models in the hypothesis space $\mathcal{H}$. Let $\theta_u \sim U(D, z, A(D))$ denote an unlearned model, and let $\theta_r \sim A(D \setminus \{z\})$ denote a retrained model obtained by the dataset $D$ with $z$ removed.*

*The mechanism $U$ satisfies ($\varepsilon, \delta, \kappa$)-Suppressive Machine Unlearning if for a given dataset $D$ and any $z \in D$, the following conditions hold:*

1. $(\varepsilon, \delta)$-**Data Removal Condition.** *For any measurable set $T \subseteq \mathcal{H}$,*

$$\Pr\big(U(D, z, A(D)) \in T\big) \ \leq \ e^\varepsilon \Pr\big(A(D \setminus \{z\}) \in T\big) + \delta,$$

*and*

$$\Pr\big(A(D \setminus \{z\}) \in T\big) \ \leq \ e^\varepsilon \Pr\big(U(D, z, A(D)) \in T\big) + \delta. \tag{4}$$

2. ***Knowledge Suppression Condition ($\kappa$-Suppression).*** *Let $\mathbb{Q}_f$ denote the forget-target distribution and $\mathbb{Q}_0$ a reference distribution. For a given score function $s : \mathcal{X} \times \mathcal{Y} \to [0, 1]$, the suppression functional $g_s(\theta)$ for a model $\theta$ is defined as*

$$g_s(\theta) \ := \ \mathbb{E}_{x_0 \sim \mathbb{Q}_0, \, y_0 \sim P_\theta(\cdot|x_0)} s(x_0, y_0) \ - \ \mathbb{E}_{x_f \sim \mathbb{Q}_f, \, y_f \sim P_\theta(\cdot|x_f)} s(x_f, y_f). \tag{5}$$

*Then, $U$ satisfies $\kappa$-Suppression if*

$$\mathbb{E}_{\theta_u \sim U(D, z, A(D))}\Big[g_s(\theta_u)\Big] \ \geq \ \kappa. \tag{6}$$

Our $(\varepsilon, \delta, \kappa)$-Suppressive Machine Unlearning couples a mechanism-level guarantee for deletion and an operational target for suppression. The first condition, $(\varepsilon, \delta)$-data removal, provides the formal deletion guarantee ensuring that the unlearned model distribution is statistically indistinguishable from that of a model retrained on $D \setminus \{z\}$. The second, $\kappa$-Suppression, specifies the operational target, requiring the (possibly randomized) unlearning mechanism $U$ to produce a model that exhibits an expected performance gap of at least $\kappa$ between a reference distribution $\mathbb{Q}_0$ and the forget-target distribution $\mathbb{Q}_f$, as measured by a score function $s$. Consequently, these conditions jointly regulate the underlying statistical properties of the model after unlearning and its functional behavior with respect to the target knowledge.

### 4.1 THEORETICAL ANALYSIS

This section provides the theoretical analysis for Suppressive Machine Unlearning. We aim to analyze the relationship between the mechanism-level guarantee of $(\varepsilon, \delta)$-Data Removal and the operational objective of $\kappa$-Suppresion. The analysis proceeds by first extending the data removal guarantee to a batch/sequential setting involving the removal of multiple data points, establishing a composition theorem (Lemma 4.2). Then, we demonstrate that this group-level distributional guarantee implies bounds on the operational objective of $\kappa$-Suppression (Theorem 4.4). All proofs are deferred to Appendix A.

To facilitate the analysis, we define probability measures $\mu, \nu$ on $\mathcal{H}$ by

$$\mu(T) := \Pr\big(U(D, z, A(D)) \in T\big), \qquad \nu(T) := \Pr\big(A(D \setminus \{z\}) \in T\big), \tag{7}$$

for all measurable $T \subseteq \mathcal{H}$. The $(\varepsilon, \delta)$-Data Removal Condition (Eq. 4) holds iff for all measurable $T \subseteq \mathcal{H}$, $\mu(T) \leq e^\varepsilon \nu(T) + \delta$ and $\nu(T) \leq e^\varepsilon \mu(T) + \delta$.

**Lemma 4.2** (Group-Data Removal for $D_f$). *Assume an unlearning mechanism $U$ satisfies the $(\varepsilon, \delta)$-Data Removal Condition (Definition 2.2) for any dataset $D$ and any $z \in D$. Let $D_f = \{z_1, \ldots, z_k\} \subseteq D$ be a forget set of $k \geq 1$ points.*

*Define a sequence of datasets $S_0 := D$ and $S_i := D \setminus \{z_1, \ldots, z_i\}$ for $i = 1, \ldots, k$. Consider a sequence of random models generated by the following process:*

1. *Let $\Theta_0 \sim A(D)$ be the initial model trained on $D$.*

2. *For $i = 1 \ldots, k$, let $\Theta_i \sim U(S_{i-1}, z_i, \Theta_{i-1})$ be the model obtained after unlearning $z_i$.*

*Let $\mu_i$ and $\nu_i$ denote the probability measures on $\mathcal{H}$ induced by $\Theta_i$ and $A(S_i)$, respectively, i.e.,*

$$\mu_i(T) \ := \ \Pr(\Theta_i \in T) \quad \text{and} \quad \nu_i(T) \ := \ \Pr\big(A(S_i) \in T\big) \quad \text{for } T \subseteq \mathcal{H}.$$

*Then, for every measurable $T \subseteq \mathcal{H}$, the following holds:*

$$\mu_k(T) \ \leq \ e^{k\varepsilon} \, \nu_k(T) \ + \ \delta \sum_{j=0}^{k-1} e^{j\varepsilon} \quad \text{and} \quad \nu_k(T) \ \leq \ e^{k\varepsilon} \, \mu_k(T) \ + \ \delta \sum_{j=0}^{k-1} e^{j\varepsilon}. \tag{8}$$

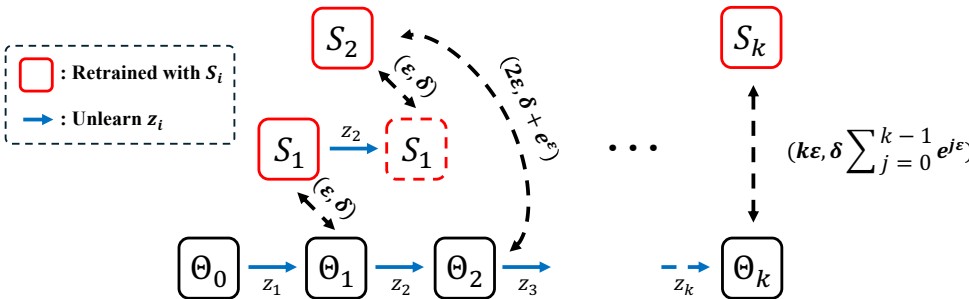

Figure 1: Illustration of Lemma 4.2 (Group-Data Removal). Starting from an initial model trained on dataset $D$, each unlearning step $U$ removes an element $z_i \in D_f$. The process yield the cumulative removal parameters between the distribution of unlearned models and that of retraining on $D \setminus D_f$.

Lemma 4.2 shows that the data removal guarantee composes over a sequence of unlearning operations. We formally define that an unlearning mechanism $U$ satisfies $(\varepsilon_k, \delta_k)$-Group-Data Removal for a forget set $D_f$ if the final distributions $\mu_k$ and $\nu_k$ are $(\varepsilon_k, \delta_k)$-indistinguishable. In particular, Lemma 4.2 implies $(k\varepsilon, \delta \sum_{j=0}^{k-1} e^{j\varepsilon})$-group data removal. This result extends to the adaptive setting, where the parameters $(\varepsilon^{(i)}, \delta^{(i)})$ may vary across steps. Fig. 1 illustrates this lemma.

**Corollary 4.3** (Adaptive Group-Data Removal). *Under Lemma 4.2, suppose that at step $i$ the guarantee of unlearning mechanism holds with parameters $(\varepsilon^{(i)}, \delta^{(i)})$, possibly depending on $(S_{i-1}, z_i)$. Then, for all $T \subseteq \mathcal{H}$,*

$$\mu_k(T) \ \leq \ e^{\sum_{i=1}^{k} \varepsilon^{(i)}} \nu_k(T) \ + \ \sum_{i=1}^{k} e^{\sum_{t=1}^{i-1} \varepsilon^{(t)}} \delta^{(i)}, \tag{9}$$

*and*

$$\nu_k(T) \ \leq \ e^{\sum_{i=1}^{k} \varepsilon^{(i)}} \mu_k(T) \ + \ \sum_{i=1}^{k} e^{\sum_{t=i+1}^{k} \varepsilon^{(t)}} \delta^{(i)}. \tag{10}$$

In the adaptive case, $\varepsilon$ adds linearly, while the $\delta$ accumulates with exponential weights, making the final guarantee dependent on the removal order. Note that when $(\varepsilon^{(i)}, \delta^{(i)}) \equiv (\varepsilon, \delta)$ for all $i$, these bounds reduce to the non-adaptive case in Eq. 8 that is order-independent.

Lemma 4.2 and Corollary 4.3 establish a composition theorem for unlearning, which is directly analogous to the group privacy principle in differential privacy (Dwork et al., 2014). An approximate removal per item implies distributional closeness after forgetting any set of items, with parameters $\varepsilon$ and $\delta$ accumulated similarly as in the group privacy. This connection helps us to leverage privacy accounting intuitions for unlearning sequences. Lemma 4.2 formalizes unlearning requests that arrive sequentially in a stream, a general case that inherently covers batched requests. If an implementation chooses $(\varepsilon^{(i)}, \delta^{(i)})$ adaptively depending on $z_i$, then the sequential composition becomes order-dependent with $(\varepsilon_k, \delta_k) = (\sum_{i=1}^{k} \varepsilon^{(i)}, \sum_{i=1}^{k} e^{\sum_{t=1}^{i-1} \varepsilon^{(t)}} \delta^{(i)})$ (Eq. 9).

Building on this composition, we now translate distributional closeness into guarantees on the operational suppression. We show that Group-Data Removal with parameters $(\varepsilon_k, \delta_k)$ yields the bounds relating the unlearned model's suppression $\kappa_u$ to the retrained baseline $\kappa_r$.

**Theorem 4.4** (Suppression Transfer under Group-Data Removal). *Let an unlearning mechanism satisfy $(\varepsilon_k, \delta_k)$-Group Data Removal for the forget set $D_f$ of size $k \geq 1$. Consider the suppression functional $g_s(\theta)$ derived from a bounded score function $s : \mathcal{X} \times \mathcal{Y} \to [0, 1]$.*

*Let $\kappa_u$ and $\kappa_r$ be the suppression metrics for the unlearned and retrained models, respectively:*

$$\kappa_u := \mathbb{E}_{\theta_u \sim U(D, D_f, A(D))}[g_s(\theta_u)], \qquad \kappa_r := \mathbb{E}_{\theta_r \sim A(D \setminus D_f)}[g_s(\theta_r)].$$

*Then,*

$$|\kappa_u - \kappa_r| \ \leq \ 2\big(e^{\varepsilon_k} - 1\big) \ + \ 2\delta_k \tag{11}$$

*and, more sharply (one–sided),*

$$\boxed{\kappa_u \ \leq \ e^{\varepsilon_k}\kappa_r \ + \ 2\delta_k + e^{\varepsilon_k} - 1 = \mathcal{O}(e^{\varepsilon_k}\kappa_r)} \tag{12}$$

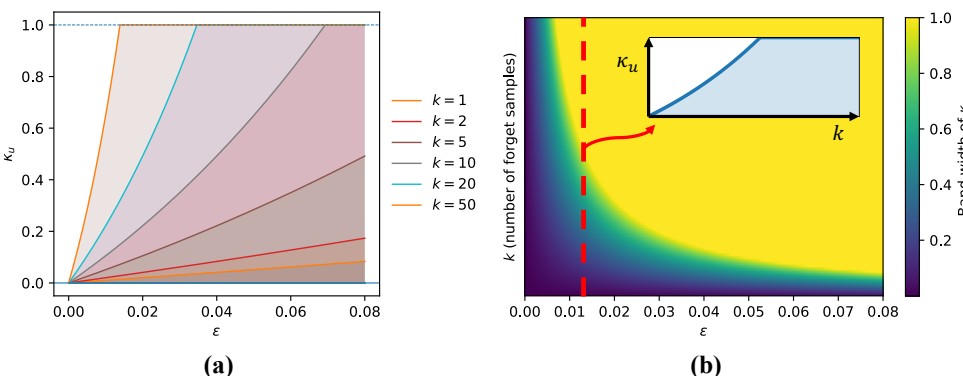

Figure 2: **(a)** Impact of $\varepsilon$ on suppression bounds $\kappa_u$. This plot shows the upper bounds of the suppression level $\kappa_u$ as a function of the removal parameter $\varepsilon$ at each $k$. Shaded bands mark the feasible range of $\kappa_u$. **(b)** Heatmap of the width of the feasible interval for $\kappa_u$ over the $(\varepsilon, k)$ grid. For each data removal level $\varepsilon$, we compose group-data removal guarantees for a deletion set of size $k$ to obtain $(\varepsilon_k, \delta_k)$. These are plugged into the one-sided bounds to compute upper bounds on $|\kappa_u|$. For both figures, $\kappa_r$ (retrain baseline) is fixed to 0 and $\delta$ is fixed to $10^{-5}$.

Theorem 4.4 establishes that the suppression level $\kappa_u$ achieved by an unlearned model remains coupled to the retraining baseline $\kappa_r$, with deviations bounded by $(\varepsilon_k, \delta_k)$. This validates Definition 4.1: a mechanism that both (i) satisfies group-level deletion guarantees and (ii) enforces $\kappa$-suppression behaves, on the operational metric $g_s$, almost indistinguishably from full retraining on $D \setminus D_f$. This result connects the two primary desiderata in unlearning—a mechanism-level deletion guarantee and a behavior-level obligation-within one auditable notion. Moreover, these guarantees compose across multiple unlearning requests (via Lemma 4.2), making the framework applicable to practical streaming or batched unlearning.

The one-sided bound (Eq. 12) provides a key insight into the behavior of approximate unlearning algorithms: as the deletion guarantee is relaxed (i.e., larger $\varepsilon_k$) or the forget set size ($k$) increases, the allowable deviation between $\kappa_u$ and $\kappa_r$ necessarily increases, meaning a much room for suppression. Figure 2-(b) illustrates how the feasible interval for $\kappa_u$ expands with $\varepsilon$ and the group size $k$, offering intuition for the empirical observation that most unlearning methods can exhibit a high level of suppression on the forget data when forgetting a large number of samples. Moreover, in the case of exact unlearning, the result coincides perfectly with the prevailing intuition.

**Corollary 4.5** (Suppression Parity of Exact Unlearning). *If $U$ satisfies $(\varepsilon, \delta) = (0, 0)$, for all $s$,*

$$\mathbb{E}_{\theta_u}[g_s(\theta_u)] = \mathbb{E}_{\theta_r}[g_s(\theta_r)], \qquad \kappa_u = \kappa_r. \tag{13}$$

*Consequently, for any $\kappa \geq 0$, $U$ is $\kappa$-suppressive iff retraining on $D \setminus D_f$ is $\kappa$-suppressive.*

Through our theoretical analysis, we obtain a comprehensive understanding of existing unlearning approaches with a consistent definition that couples deletion-level guarantees with behavioral suppression targets. This joint perspective enables us to reinterpret diverse practical unlearning demands—ranging from strict data-deletion requests to task-specific suppression requirements—within a common taxonomy.

### 4.2 CONNECTION TO UNLEARNING REQUESTS

**Recall.** *Alice (Data Owner) asks Bob (Model Owner) to "Erase my data from the model."* Using our notation, Alice may specify a policy target $\kappa^* \geq 0$ ("suppress at least this much on my data relative to the model's general behavior"). Three request types, now fully instantiated:

1. **Type I Request:** Require the $(\varepsilon, \delta)$-Data Removal Condition only; no extra suppression target beyond whatever the retrain achieves. Exact unlearning (Corollary 4.5) will be the gold-standard of this request. With small $(\varepsilon, \delta)$, Theorem 4.4 implies $|\kappa_u - \kappa_r|$ is tightly bounded, so $U$ tracks the retrain on the operational metric. This explains the conventional evaluation framework for producing an unlearning model whose performance is as close as possible to that of a retrained model.

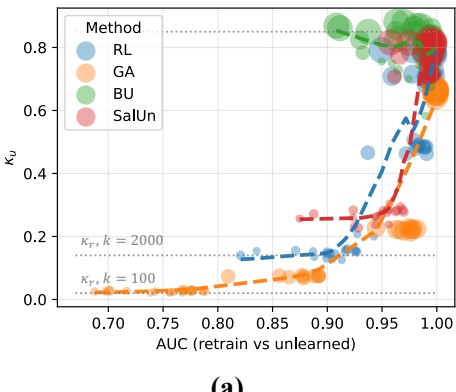 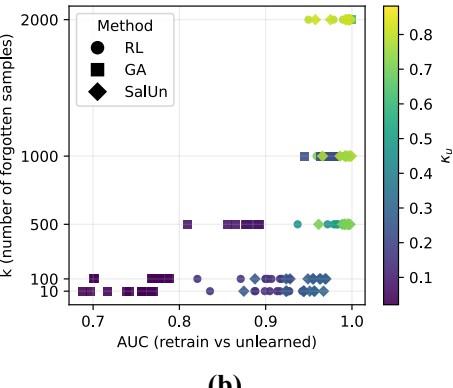

**(a)**                                             **(b)**

Figure 3: **(a)** Suppression level across unlearning methods. Point size encodes the number of forget data $k$, and color-dotted lines denote per-method trend lines. Retrained models' suppression $\kappa_r$ was represented by gray-dotted lines (only representative figures are provided). **(b)** Indistinguishability (shown by AUC) vs. forget-set size via heatmap-style scatter. Point color encodes the achieved suppression level $\kappa_u$.

2. **Type II Request:** Impose both conditions in Definition 4.1: the $(\varepsilon, \delta)$-Data Removal Condition and the $\kappa^*$-suppression constraint. The maximum achievable $\kappa_u$ consistent with $(\varepsilon_k, \delta_k)$ is upper-bounded by Eq. 12. Hence a necessary feasibility condition for any target $\kappa^*$ is $\kappa^* \leq e^{\varepsilon_k}\kappa_r + 2\delta_k + e^{\varepsilon_k} - 1$ (immediate from the one-sided bound). Equivalently, the minimum deletion budget that can support $\kappa_u$ is

$$\varepsilon_k \geq \log\frac{\kappa_u + 1 - 2\delta_k}{\kappa_r + 1}. \tag{14}$$

Thus, stronger suppression targets force looser approximate unlearning guarantees.

3. **Type III Request:** No claim about deletion; only require $\kappa_u \geq \kappa^*$ with capability preserved on $\mathbb{Q}_0$. This aligns with recent suppression–centric methods: it lives on the behavioral axis without constraining parameter-level deletion. This request falls outside the typical definition of unlearning: these notions do not provide a connection between removal and suppression. Our definition does—by directly enforcing auditable $\kappa$-suppression.

In conclusion, our framework is request-complete: it uniformly covers **Type I–III requests**, by specifying behavioral targets on outputs. Taken together, these results position our definition as both principled and practical for real-world unlearning deployments.

### 4.3 EMPIRICAL ANALYSIS

We introduced $(\varepsilon, \delta, \kappa)$-Suppressive Machine Unlearning, which couples data removal guarantees with performance suppression, and show that the suppression level $\kappa_u$ is controlled by $(\varepsilon_k, \delta_k)$ and $\kappa_r$. We now examine how these theoretical constraints manifest in practical unlearning pipelines. Our core idea is to jointly observe the indistinguishability among retrained models and unlearned models along with their suppression level $\kappa_u$. In typical unlearning pipelines, it is practically infeasible to estimate the removal parameters $(\varepsilon, \delta)$: unlearning operators are not standardized DP mechanisms, any $(\varepsilon, \delta)$ accounting would be highly method-specific. Motivated by (Ghazi & Issa, 2024), to obtain a method-agnostic, observable surrogate for "how distinguishable" an unlearned model is from its retrain counterpart, we therefore employ a logistic discriminator trained on soft logits to compute the ROC–AUC; lower AUC indicates reduced distinguishability—operationally aligning with a smaller $\varepsilon$—while higher AUC indicates the opposite.

**Setting.** We utilize CIFAR-10 and a ResNet-18 backbone. The forget dataset $D_f$ comprises $k \in \{50, 100, 500, 1000, 2000\}$ training samples from class 0. The retrain baselines for each case train from scratch on $D \setminus D_f$. We perform unlearning with 10 different random seeds under exactly the same conditions and unlearning mechanisms across all unlearning methods, regardless of $k$. We test four unlearning pipelines to remove/suppress $D_f$: (1) Random Labeling (GA) (Golatkar et al.,

2020), (2) Gradient Ascent (GA) (Thudi et al., 2022a), (3) Boundary Unlearning (BU) (Chen et al., 2023), and (4) Saliency Unlearning (SalUn) (Fan et al., 2023). The reference distribution $\mathbb{Q}_0$ is the entire test set excluding class 0 and the forget-target distribution $\mathbb{Q}_f$ is the class 0-only test set. The score function $s$ is a classification accuracy, so the suppression level $\kappa$ is $Acc(\mathbb{Q}_0) - Acc(\mathbb{Q}_f)$.

**Results.** We scatter various unlearned models with varying $k$ on AUC-$\kappa_u$ space, intended to surrogate $\varepsilon$-$\kappa$ space from Fig. 2-(a). As shown in Fig. 3-(a), the curves follow an exponential trend consistent with the one-sided bound established in Eq. 12 ($\kappa_u \leq \mathcal{O}(e^{\varepsilon_k}\kappa_r)$). For intuition, note that enlarging either the deletion budget $\varepsilon_k$ widens the feasible gap between $\kappa_u$ and $\kappa_r$. However, in the case of Boundary Unlearning (BU), it only behaves in the right-upper part of the trend. Because it operates by collapsing the decision boundary of a specific class irrespective of the unlearning setting, it consistently achieves a high level of suppression. In Fig. 3-(b), we scatter each unlearned model on AUC-$k$ space, which corresponds to $\varepsilon$-$k$ space in its theoretical counterpart, with a color map for representing the suppression level $\kappa_u$. Our result exhibits a trend consistent with the heatmap in Fig. 2-(b), where dark blue-colored points locate the left or lower region, while yellow-colored points scatter on the right-upper area. We also observe a structural trade-off implied by the composition of group-data removal: unlearning accumulates $(\varepsilon, \delta)$ in a way that mirrors group privacy, which explains the joint movement of indistinguishability and suppression as $k$ increases.

Taken together—(i) the composition-driven accumulation of $(\varepsilon, \delta)$, (ii) the suppression transfer bound of Theorem 4.4, and (iii) the AUC-based indistinguishability surrogate—our results substantiate that $(\varepsilon, \delta, \kappa)$-Suppressive Machine Unlearning coherently couples verifiable deletion with operational suppression in practical pipelines.

## 5 DISCUSSION

**Challenges in Empirical Validation for LLMs.** Unlearning in the LLMs is the most active area of discussion on the suppression-perspective. While our definitions and theorems are model-agnostic, its empirical validation on LLMs raises practical challenges. First, our evaluation requires multiple independent retraining runs to estimate the distribution of the retrained models. Full retraining of such models is prohibitive due to their computational cost and restricted access to large-scale training corpora. Second, measuring how deletion/suppression varies with a forget set size $k$ is impractical for LLMs because clear trends may only emerge at an exponential scale, a requirement compounded by the difficulty of accessing such vast datasets. Second, measuring how deletion/suppression vary with a forget set size $k$ is not auditable in the same way as our setting.

**Suppression (Type III) as Pragmatic Unlearning.** We argue that suppression-only requests (Type III) can be recognized as a legitimate form of unlearning, particularly in scenarios where curated forget sets and multiple retrains are infeasible. In such cases, auditable and policy-defined $\kappa$-suppression on a target distribution offers an guarantee of reduced capability, complementing data-removal baselines rather than replacing them. We do not claim that suppression certifies deletion; rather, it establishes a measurable behavioral contract when data provenance is uncontrollable, making our $(\epsilon, \delta, \kappa)$ definition operationally meaningful for large models. Nevertheless, the scope of "unlearning" under output-level criteria remains an active area of debate. For instance, relearning attacks (Hu et al., 2024; Fan et al., 2025) reveal that naïve refusal/suppression can be reversible.

## 6 CONCLUSION

Machine unlearning has emerged as a critical capability for modern AI systems, yet existing definitions have been insufficient to capture the full spectrum of real-world unlearning demands. This work addresses a fundamental gap in the field by introducing Suppressive Machine Unlearning, a unified framework that encompasses both data removal and knowledge suppression objectives under a single, theoretically grounded definition. The implications of this work extend beyond theoretical interest. As AI systems become more pervasive and face increasing scrutiny regarding their training data and capabilities, the ability to audit both what has been removed and what has been suppressed becomes essential. Our $(\varepsilon, \delta, \kappa)$ framework provides the mathematical foundation for such auditing, enabling organizations to make principled decisions about unlearning trade-offs while meeting diverse stakeholder demands.

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

# A   PROOFS OF THEORETICAL ANALYSIS

## A.1   PROOF OF LEMMA 4.2.

We prove the forward inequality; the reverse direction is identical by symmetry.

**Setup and notation**   $\mathcal{H}$ denotes the model (parameter) space; we view it as a standard Borel space. Its $\sigma$-algebra of measurable sets is $\mathcal{B}(\mathcal{H})$. We write $T \in \mathcal{B}(\mathcal{H})$ for measurable "events" about models. $A$ is a (possibly randomized) learner. For a dataset $S$, the random model $A(S) \in \mathcal{H}$ induces the distribution $\nu_S(T) := \Pr(A(S) \in T)$ on $\mathcal{H}$. $U$ is a randomized update that takes a dataset $S$ containing $z$, a current model $h \in \mathcal{H}$, and internal randomness, and outputs a new model in $\mathcal{H}$. Given $S$ and $z$, the mapping $h \mapsto U(S, z, h)$ is measurable and randomized. As in the lemma, $S_0 := D$ and $S_i := D \setminus \{z_1, \dots, z_i\}$. We set $\Theta_0 \sim A(D)$ and for $i \geq 1$ sample $\Theta_i \sim U(S_{i-1}, z_i, \Theta_{i-1})$. Let $\mu_i(T) := \Pr(\Theta_i \in T)$ and $\nu_i(T) := \Pr(A(S_i) \in T)$.

**Why we may treat $U$ as a "Markov kernel."**   Fix $(S_{i-1}, z_i)$. Because $U$ is a randomized algorithm that is measurable in its inputs, there exists a setwise mapping

$$K_i : \mathcal{H} \times \mathcal{B}(\mathcal{H}) \to [0, 1], \qquad K_i(h, T) := \Pr\big(U(S_{i-1}, z_i, h) \in T\big),$$

such that: (i) for each $h$, $T \mapsto K_i(h, T)$ is a probability measure on $\mathcal{B}(\mathcal{H})$; (ii) for each measurable $T$, $h \mapsto K_i(h, T)$ is measurable. These two properties are exactly what is needed to apply standard *post-processing* inequalities; With this notation we have

$$\mu_i = (\mu_{i-1} K_i)(T) := \int K_i(h, T)\, \mu_{i-1}(\mathrm{d}h) \quad \text{and} \quad (\nu_{i-1} K_i)(T) := \int K_i(h, T)\, \nu_{i-1}(\mathrm{d}h).$$

**One-step guarantee.**   Applying the single-point approximate unlearning guarantee at $(S_{i-1}, z_i)$ gives, for all measurable $T$,

$$(\nu_{i-1} K_i)(T) \leq e^\varepsilon \nu_i(T) + \delta. \tag{15}$$

**Post-processing fact (used repeatedly).**   If measures $\alpha, \beta$ on $\mathcal{H}$ satisfy $\alpha(T) \leq e^\rho \beta(T) + \eta$ for all $T \in \mathcal{B}(\mathcal{H})$, then for any kernel $K$ and all measurable $T$,

$$(\alpha K)(T) \leq e^\rho (\beta K)(T) + \eta.$$

*Reason.* The premise is equivalent (by layer-cake / indicator approximation) to $\int g\, \mathrm{d}\alpha \leq e^\rho \int g\, \mathrm{d}\beta + \eta$ for all measurable $g : \mathcal{H} \to [0, 1]$. Taking $g(h) = K(h, T)$ yields the claim.

**Inductive invariant.**   We prove by induction on $i$ that for all measurable $T$,

$$\mu_i(T) \leq e^{i\varepsilon} \nu_i(T) + \delta \sum_{j=0}^{i-1} e^{j\varepsilon} \qquad (*_i) \tag{16}$$

*Base case $i = 1$.* By Definition 2.2 at $(S_0, z_1)$ and the fact $\mu_1 = \nu_0 K_1$,

$$\mu_1(T) = (\nu_0 K_1)(T) \leq e^\varepsilon \nu_1(T) + \delta,$$

which is $(*_1)$.

*Inductive step.* Assume $(*_{i-1})$. Applying the same kernel $K_i$ to both sides and using the post-processing fact,

$$\mu_i(T) = (\mu_{i-1} K_i)(T) \leq e^{(i-1)\varepsilon} (\nu_{i-1} K_i)(T) + \delta \sum_{j=0}^{i-2} e^{j\varepsilon}.$$

Combine this with the one-step bound Eq. 15 to get

$$\mu_i(T) \leq e^{(i-1)\varepsilon} \big(e^\varepsilon \nu_i(T) + \delta\big) + \delta \sum_{j=0}^{i-2} e^{j\varepsilon} = e^{i\varepsilon} \nu_i(T) + \delta \sum_{j=0}^{i-1} e^{j\varepsilon},$$

which is $(*_i)$. Taking $i = k$ yields the forward inequality of the lemma. The reverse inequality follows by applying the symmetric direction in Definition 2.2 at each step (swap the roles of $\mu$ and $\nu$). $\qquad \square$

## A.2 PROOF OF THEOREM 4.4.

*Proof.* Since $s \in [0, 1]$, it follows that $g_s \in [-1, 1]$, hence $|\kappa_u| \leq 1$ and $|\kappa_r| \leq 1$. For simplicity, set $\varepsilon := \varepsilon_k$ and $\delta := \delta_k$.

**Step 1 (from setwise bounds to bounds on integrals).** Fix any measurable $f : \mathcal{H} \to [0, 1]$. For $t \in [0, 1]$ let $T_t := \{\theta \in \mathcal{H} : f(\theta) \geq t\}$. By the layer-cake representation and Tonelli's theorem,

$$\int f \, d\mu = \int_0^1 \mu(T_t) \, dt, \qquad \int f \, d\nu = \int_0^1 \nu(T_t) \, dt. \tag{17}$$

Applying $\mu(T_t) \leq e^\varepsilon \nu(T_t) + \delta$ pointwise in $t$ and integrating over $t \in [0, 1]$ yields

$$\int f \, d\mu \leq e^\varepsilon \int f \, d\nu + \int_0^1 \delta \, dt = e^\varepsilon \int f \, d\nu + \delta. \tag{18}$$

Similarly, from $\nu(T_t) \leq e^\varepsilon \mu(T_t) + \delta$ we obtain

$$\int f \, d\nu \leq e^\varepsilon \int f \, d\mu + \delta. \tag{19}$$

**Step 2 (one–sided bounds).** Let $f := (g_s + 1)/2$, which maps $\mathcal{H}$ into $[0, 1]$. Then

$$\kappa_u = \int g_s \, d\mu = 2 \int f \, d\mu - 1, \qquad \kappa_r = \int g_s \, d\nu = 2 \int f \, d\nu - 1. \tag{20}$$

Using $\int f \, d\mu \leq e^\varepsilon \int f \, d\nu + \delta$ gives

$$\kappa_u = 2 \int f \, d\mu - 1 \leq 2\left(e^\varepsilon \int f \, d\nu + \delta\right) - 1 = e^\varepsilon(\kappa_r + 1) + 2\delta - 1 = e^\varepsilon \kappa_r + (e^\varepsilon - 1) + 2\delta. \tag{21}$$

From $\int f \, d\nu \leq e^\varepsilon \int f \, d\mu + \delta$ we obtain

$$\int f \, d\mu \geq e^{-\varepsilon} \int f \, d\nu - e^{-\varepsilon} \delta, \tag{22}$$

hence

$$\kappa_u = 2 \int f \, d\mu - 1 \geq 2e^{-\varepsilon} \int f \, d\nu - 2e^{-\varepsilon} \delta - 1$$
$$= e^{-\varepsilon}(\kappa_r + 1) - 2e^{-\varepsilon}\delta - 1 = e^{-\varepsilon}\kappa_r - e^{-\varepsilon}(2\delta + e^\varepsilon - 1). \tag{23}$$

The inequalities Eq. 21 and Eq. 23 are the claimed one–sided bounds.

**Step 3 (symmetric bound on $|\kappa_u - \kappa_r|$).** From Eq. 21,

$$\kappa_u - \kappa_r \leq (e^\varepsilon - 1)\kappa_r + (e^\varepsilon - 1) + 2\delta \leq 2(e^\varepsilon - 1) + 2\delta, \tag{24}$$

since $|\kappa_r| \leq 1$.

Exchanging the roles of $\mu$ and $\nu$ in Eq. 21 gives $\kappa_r \leq e^\varepsilon \kappa_u + (e^\varepsilon - 1) + 2\delta$, hence

$$\kappa_u - \kappa_r \geq \kappa_u - \left(e^\varepsilon \kappa_u + (e^\varepsilon - 1) + 2\delta\right) = (1 - e^\varepsilon)\kappa_u - (e^\varepsilon - 1) - 2\delta \geq -2(e^\varepsilon - 1) - 2\delta, \tag{25}$$

since $|\kappa_u| \leq 1$.

Combining the two displays yields $|\kappa_u - \kappa_r| \leq 2(e^\varepsilon - 1) + 2\delta$. $\qquad \square$

# B EXPERIMENTAL DETAILS

## B.1 ALGORITHM DESCRIPTIONS

Here we present the detailed description of each unlearning pipeline. (1) Random Labeling (RL) (Golatkar et al., 2020): Replace the forget-set labels with random ones and briefly retrain so their learning signal becomes noise, eroding memorization. (2) Gradient Ascent (GA) (Thudi et al., 2022a): Fine-tune on the forget data by ascending the loss, actively pushing parameters away from fitting those examples. (3) Boundary Unlearning (BU) (Chen et al., 2023): Shrink the margin around forget samples to retract the decision boundary and lower confidence in that region. (4) Saliency Unlearning (SalUn) (Fan et al., 2023): Identify parameters salient to the forget set via gradient/saliency scores, selectively reset or weaken them, then briefly fine-tune on the retain data to restore overall performance.

## B.2   CONFIGURATION

All experiments performed unlearning with fixed hyperparameters according to each unlearning method on 10 random seeds.

For orginal and retrained models, we trained ResNet-18 classifiers on CIFAR-10 (32×32 resolution). During training, we applied RandomCrop (padding=4) and RandomHorizontalFlip for data augmentation. Inputs were normalized using ImageNet statistics. Optimization used SGD with an initial learning rate of 0.1, momentum 0.9, and weight decay of 0.0005, for a total of 200 epochs. A MultiStepLR schedule reduced the learning rate by a factor of 0.2 at epochs 60, 120, and 160. The training batch size was 256; evaluation used a batch size of 100. We used cross-entropy loss and reported top-1 accuracy.

In Random Labeling scenarios, forget data and randomly selected labels were trained in pairs. Optimization used SGD with a learning rate of 0.0002. In Gradient Ascent scenarios, the model was updated by making the loss calculated for the forget data negative. Optimization used AdamW with a learning rate of $2 \times 10^{-6}$. In Boundary Unlearning scenarios, adversarial examples are generated on-the-fly using an FGSM agent configured with an $\ell_\infty$ step bound of 0.3, random initialization enabled. And we find adjacent classes for adversarial samples with an original model. We optimize it with SGD at a small constant learning rate 0.0005. For Saliency Unlearning, we use the unlearning data to accumulate gradients of the negative cross-entropy $-\mathrm{CE}(f(x), y)$, take absolute values, rank all parameters globally, and keep the top 50% as a binary mask; we then train a fresh copy of the model for one epoch on the same data using random labels drawn uniformly from the remaining classes, SGD ($\mathrm{lr} = 3 \times 10^{-4}$), and multiply each parameter gradient by the mask so only selected entries update.

