# OpenReview forum: "Rethinking the Definition of Unlearning: Suppressive Machine Unlearning"
_ICLR.cc/2026/Conference — ICLR 2026 Conference Withdrawn Submission_

### Official Review · Reviewer_G3Ui · 2025-10-29

**Soundness:** 2
**Presentation:** 3
**Contribution:** 1
**Rating:** 2
**Confidence:** 4

**Summary:**

This paper proposes a new definition for machine unlearning called "Suppressive Machine Unlearning." This definition aims to formally capture two aspects: 1) the proximity to a full retrain, similar to ($\varepsilon$, $\delta$)-approximate unlearning, and 2) a new "suppression" parameter, $\kappa$, which quantifies the reduction in the model's utility on the specific data to be forgotten. The authors present a theoretical analysis of the trade-off between these components and provide experimental results to support their framework.

**Strengths:**

* The paper introduces a new definition, "Suppressive Machine Unlearning," which is a reasonable goal. Attempting to formally unify the concepts of removal (proximity to retraining) and suppression (utility reduction) is a relevant direction for the field.

* The discussion in lines 466-473 regarding failure modes is interesting. Distinguishing between true data suppression and a general loss of model capability is an important problem, and the paper correctly identifies this. This part of the discussion is a good starting point for a more in-depth analysis.

**Weaknesses:**

Weaknesses below are ordered more or less following importance:

1. **Questionable Novelty.** The paper's core framing appears to have significant overlap with prior work, particularly the omitted paper "Distributional Machine Unlearning via Selective Data Removal" by Allouah et al. (ICML'25 Machine Unlearning for Generative AI). That paper seems to formalize and quantify several of the key ideas presented here, such as utility on a target distribution (Def 3.1 in this paper vs. Prop. 2 in the omitted paper). The conceptual and technical novelty of this work, when compared to this relevant paper, is not clear.

2. **Potentially Vacuous Bounds.** The main theoretical result, Theorem 4.4, and its consequences (e.g., Eq. 14) may be vacuous in practice. To maintain any reasonable utility, DP-based methods often require an ε on the order of 10 or more. In this regime, the bound given (which is exponential in ε₁) would become extremely large, offering no meaningful guarantee even for a single data point deletion, let alone multiple.

3.1 **Limited Theoretical Novelty.** Several of the theoretical results seem to be reformulations of existing concepts. For instance, Lemma 4.2 and Corollary 4.3 are essentially restatements of known group privacy results in differential privacy. While the authors allude to this, it would be more transparent to directly cite the relevant group privacy theorem rather than presenting it as two separate results plus a figure.

3.2 **Limited Empirical Validation.** The experiments are not validated in settings where unlearning guarantees (like ε) can be computed exactly. Testing on convex models, e.g., Guo et al. (2019) or specific non-convex settings, e.g., Koloskova et al. (2025) "Certified Unlearning for Neural Networks", where certified unlearning is possible would provide a much stronger and clearer validation of the paper's theoretical claims, beyond empirical suppression scores.

4. **Misleading Experimental Interpretation.** The interpretation of Figure 3-a, which claims to verify an exponential trend, might be misleading. The observed trend could simply be an artifact of increasing the number of forgotten samples (k). Looking at a fixed k, there is no obvious trend; the suppression level appears to be more or less constant.

5. **Framing of the κ Parameter.** The value of adding κ as part of the mechanism definition is unclear. It seems more natural to target ε-certification, and as Theorem 4.4 itself shows, this certification already implies a corresponding κ value. The current framing feels a bit redundant.

6. **Insufficient Discussion on Failure Modes.** While the paper touches on an interesting point (lines 466-473), the discussion on failure modes is far too brief. For example, one way to improve the discussion is the following failure mode: if an unlearning operation just damages the model's overall capabilities, it might appear to have "suppressed" the target data (by having low utility everywhere), but the data may still be identifiable or recoverable. This would mean unlearning hasn't actually happened. The paper needs to engage with this possibility more deeply.

**Questions:**

In addition to the weaknesses above, here are some questions for the authors to address:

1. Novelty vs. Allouah et al.: Could the authors please elaborate on the specific novelty of their framework compared to the "Distributional Machine Unlearning via Selective Data Removal"? That work seems to be very closely related in its framing (lines 70, 148-149) and quantifies similar ideas.

2. Missing Citations: For Definitions 2.1 and 2.2, should credit be given to Ginart et al. (2019) "Making AI Forget You: Data Deletion in Machine Learning", who appear to have introduced these (or very similar) definitions earlier?

3. Missing MIA Evaluation: Why were Membership Inference Attacks (MIAs) not used in the experimental evaluation (Sec 4.3)? MIAs are a fairly standard benchmark for unlearning, and several papers have adapted them for this setting. Their omission seems like a significant gap in the evaluation.

4. Inconsistency in Definitions: Why does Definition 4.1 consider only a single forget sample, whereas Definition 2.2 considers a set? This seems inconsistent.

5. Grammatical Clarity: Some sentences are grammatically unclear, which hinders understanding. For example: "What and how constitutes such unlearning requests" (line ~53) and "We define this condition directly on the properties of a single (post-unlearning)." (line 158). Could the authors please review and rephrase these for clarity?

6. Suggestion on Verification: Have the authors considered verifying their claims in settings where ε and δ can be computed exactly, for instance, in convex models? This could provide a clearer validation of the theoretical bounds.

---

### Official Review · Reviewer_Xiht · 2025-10-29

**Soundness:** 2
**Presentation:** 2
**Contribution:** 2
**Rating:** 2
**Confidence:** 4

**Summary:**

The paper aims to provide a single unifying definition of unlearning encompassing data-removal and knowledge suppression.
Specifically, it proposes "Suppressive Machine Unlearning", a unified definition that blends (ε,δ) data unlearning with a knowledge-suppression threshold k.
The paper nicely maps real-world requests into three types: erase only (Type I), erase + suppress (Type II), and suppress only (Type III). It proves how (ε,δ) accumulate over multiple forget requests and derives a link on the achievable suppression of an unlearned model to the retrain baseline. It provides empirical results (over  ResNet-18/CIFAR-10) showing alignment with theory.

**Strengths:**

The paper is well-written and organized.
The unifying definition (ε,δ,κ), blending data removal and knowledge suppression, is interesting.
It provides a nice taxonomy mapping real-world request types (“erase,” “erase+refuse,” “refuse only”) to guarantees.
The paper provides solid formal approaches, e.g., it derives bounds tying deletion budgets to achievable suppression.
The proposal is model/task-agnostic applicable to labels, logits, embeddings, tokens, etc.
Preliminary empirical results align with theory.

**Weaknesses:**

1. It is not clear why we need a single unifying unlearning definition. As of now, researchers discuss "example-level unlearning" (what the authors here call "data removal"), and concept/entity unlearning, etc (what the authors here call knowledge unlearning (suppression)).
These definitions pertain to different application scenarios -- e.g. in discriminative models, example-level unlearning makes sense, whereas knowledge suppression arguably does not! So **why should we use a single unlearning definition that includes issues that the application domain does not care about?**

DItto for generative models, memorized-data unlearning makes sense (for copyrights etc.) and concept-level unlearning also makes sense.
But, importantly, **these are separate*problems (example-level, memorized-data unlearning vs concept unlearning)** and require different approaches and assumptions and mechanisms, differing on difficulty and complexity! And, differing also on the appropriate metrics to quantify successes!
So why should we have a single overarching unlearning definition?
The motivation and positioning, as it stands now, appears too weak/unconvincing. These significantly reduce the significant of the contributions of the paper

2. The paper **misses citing a lot of influential research since 2023** in both example-level unlearning (in discriminative models and in unlearning memorized data in LLMs or memorized prompts in T2I models, etc). I urge the authors to review the major conferences since 2023 (say in NeurIPS, ICML, ICLR) and cite at least key influential unlearning works. This will shed insights as to how data-removal and concept unlearning differ, as they should - or at least defend the paper against the idea of keeping these separate.

3. The statement in the paper "even retrained models can preserve broad generalizations that still enable inference about the forget target" in page 3, is **(i) well-known, (ii) unavoidable, and (iii) desirable**. The NeurIPS24 paper by Zhao et al, (not cited/discussed) for example, showed when this inference (accuracy on forget examples) is almost guaranteed based on memorization levels of forget examples and on overlaps in embedding space between forget and retain data. So this is not just unavoidable, but also desirable - else we lose model generalizability. Again, we need much clearer motivation for the problem studied and solutions offered.

3. Another strong concern this reviewer has with the paper is the strong possibility that it unnecessarily conflates two separate issues.
A recent paper (arxiv.org/pdf/2509.11625) shows a nice separation of concerns: **unlearning is a different problem to that of unwanted inferences at inference/test-time**. These are different problems and should be treated differently! Or at least we are better off by treating them as separate. Again, solutions require different assessments/metrics, assumptions, etc.

4. **Even if one discounts the above reservations, the empirical results are rather too simplistic (cifar10/resnet18) - so not clear if results from more complex datasets and models would still align with theory**. True, the proposal is model agnostic etc. But one can regard this as **'plausibility' versus proof of agnosticity**. So, still, one can legitimately have fundamental disbelief that results would carry over to more complex settings. Empirical results on standard LLMs may be indeed too constly to obtain, but results on (pretrained) vision transformer models (eg ViT-small or even Tiny on smaller subsets of ImageNet) are easier to obtain. Ditto for small T2I diffusion models.

5. State explicitly what would be Qf, Q0, s for LLM refusal and T2I concept suppression.

6. **In practical terms**, why would a solution to the unified problem be better than a solution to the separate problems of instance-level forgetting versus concept-level forgetting? The paper should address this explicitly. There can be settings where one of the above is required, so why use this definition? State clearly and early on that you assume an environment with mixed requests and your contribution makes sense in this setting. And given the work in arxiv.org/pdf/2509.11625 on test-time privacy, why haviing this single unifying definition helps?

**Questions:**

Please address the comments in the weakness section above.
The authors should address all weaknesses explicitly.

I remain open to improving my score if the above are addressed.

---

> ### Author Response · Authors · 2025-11-24
> **[1/2] Rebuttal**
>
> We sincerely thank the reviewers for their careful reading of our paper and for the constructive feedback. Below, we respond to the comments point by point and explain how we plan to revise the manuscript accordingly.
>
> **W1**. We fully agree with the reviewer that example-level unlearning and concept/entity unlearning differ substantially in terms of models, assumptions, mechanisms, difficulty, and evaluation metrics. Our work does not claim that these should be solved by a single algorithmic recipe. Rather, our starting point is that, despite these differences, all of these lines of work are currently published under the name “machine unlearning” in top-tier venues, while relying on heterogeneous and sometimes incompatible problem formulations. If the meaning of “machine unlearning” depends entirely on the target/application, it becomes difficult to regard it as a coherent definition at all. This observation is precisely why we argue that the definition of machine unlearning itself needs to be refined. Classical definitions rooted in the right-to-be-forgotten focus on retraining-based data removal, but do not naturally account for the growing body of knowledge-suppression/editing works that are nonetheless presented as unlearning. Our definition is intended as a step towards such a refined definition: it treats “unlearning” as a behavioral contract specified by an unlearning request (who/what should be forgotten, and to what suppression level), and then characterizes which mechanisms and guarantees are appropriate for different request types. To the best of our knowledge, prior work does not decompose unlearning requests and guarantees in this way, nor connect classical retrain-based notions and suppression-based notions under a single, policy- and auditing-oriented perspective. We will make this positioning clearer in the revised version by explicitly stating that our goal is not to collapse all unlearning problems into one operational task, but to provide a common behavioral language and definition that can encompass the diverse unlearning scenarios that the community already treats under the same name, and to clarify how their guarantees differ depending on the application.
>
> **W2**. We agree that our current citations does not fully reflect the breadth of recent work. Due to space constraints we focused on a subset of representative papers, but we acknowledge that this choice can make the landscape appear narrower than it actually is. In the revised version, we will expand the citations to better reflect these recent developments.
>
> **W3**. We agree with the reviewer that the phenomenon we mention – that a retrained model can still achieve high accuracy or useful inference on the forget target due to broad generalization – is not new. Our intention was not to claim this as a novel empirical observation, but to use it as a starting point for our motivation: precisely because retrain-based approximate unlearning can leave non-trivial utility on the forget target, recent works have begun to assume much stronger suppression requirements and still frame them as unlearning. Our work aims to bring this “stronger-than-retrain” axis inside the definition of machine unlearning. In particular, we defines knowledge suppression by comparing utility on $Q_f$ against the baseline utility (generalization) on $Q_0$. We will clarify this in the revised version by explicitly citing Zhao et al. and related analyses.
>
> **W4**. - We thank the reviewer for pointing us to Ashiq et al [1]. We see this work as starting from a motivation very close to ours: **classical retrain-equivalence unlearning is not sufficient** to capture certain realistic protection goals. In their case, this leads to test-time privacy as a separate, inference-time objective for open-weight models. Our work takes the same underlying tension in a different direction. Rather than proposing a new test-time objective, we ask how the *definition* of machine unlearning should evolve when the community already treats stronger behavioral requirements (e.g., concept/knowledge suppression) as “unlearning.” In this sense, we view test-time privacy mechanisms as complementary to our work: they address a different layer of the pipeline, whereas our contribution is to refine the training-time unlearning contract so that both retrain-equivalent data removal and stronger suppression requirements can be expressed. We will clarify this connection and separation in the revised version.
>
> [1] Ashiq, M. H., Triantafillou, P., Tseng, H. Y., & Chrysos, G. G. (2025). Inducing Uncertainty on Open-Weight Models for Test-Time Privacy in Image Recognition. arXiv preprint arXiv:2509.11625.

---

> > ### Comment · Reviewer_Xiht · 2025-11-26
> >
> > Thank you for your response.
> >
> > Regarding W1, I see your point and I am willing to accept it as reasonable.
> >
> > Regarding W2: Although I do appreciate relevant constraints, I continue to think that using a narrow set of baselines affects significantly/negatively the strength of the paper.
> >
> > Regarding W3:
> > * 1). So you agree that this 'phenomemnon' is not new and has been revealed by prior work.
> > * 2).  the claim that "recent works have begun to assume much stronger suppression requirements and still frame them as unlearning..." is not entirely correct IMO. First, some recent works (e.g. the work that first identified this phenomenon, appears to solve the problem without assuming "much stronger suppressison".. You are correct in saying that some works do. However, the fact that *some* works may be 'misguided' does not constitute a reason to view your proposal as a particular strong contribution. At least, state explicitly which wroks do this **and** which do not and why despite the existence of those that do not, your contributions/motivations remain strong.
> >
> > Regarding W4: I would need to see concrete/explicit/detailed reasoning why housing different notions of unlearning under a same umbrella definition is so important and why treating nuances as separate problems (eg instance-based unlearning vs entity/knowledge unlearning and vs unlearning vs notions like test-time privacy) that capture these differences already providing different effective solutions is not enough... I would urge the authors to consider writing a "posiiton paper" - this would be a better avenue for this type of research, perhaps.

---

> ### Author Response · Authors · 2025-11-24
> **[2/2] Rebuttal**
>
> **W5**. We agree that our current empirical setting (CIFAR-10 / ResNet-18) is relatively simple and does not by itself prove model-agnosticity. Our primary contribution is the (ε, δ, κ) definition and analysis, and we intended the experiments as a proof-of-concept illustrating the trade-offs predicted by the theory, rather than as a comprehensive large-scale benchmark. That said, we understand the concern that this can be read as plausibility rather than evidence that the framework behaves well beyond small models. In the rebuttal period, we are running additional experiments on a slightly larger setting (e.g., a ViT-style backbone or a higher-variance image dataset) to check whether the same suppression–utility patterns hold; if completed in time, we will include these results and their discussion in the revised version.
>
> **W6**.
>
> **LLM refusal.** Here we take $Q_f$ as a distribution over harmful prompts that should be refused (e.g., harmful/jailbreak datasets), and $Q_0$ as a distribution over benign prompts used to measure utility (e.g., world-knowledge, reasoning, instruction-following tasks). In line with recent LLM unlearning benchmarks, we can define $s \in [0,1]$ from existing safety/utility scores: for example, we evaluate safety via the rate of safe (1-unsafe) responses, or general utility via task accuracy and perplexity.
>
> **T2I concept suppression.** For T2I unlearning, we let $Q_f$ be the distribution over prompts that explicitly invoke the erased concept (e.g., a specific NSFW category, character, style, or logo), and $Q_0$ be a distribution over allowed prompts (generic, safe descriptions). We can evaluate (a) concept leakage via specialized detectors (NSFW/celebrity/IP classifiers or VLM-based concept detectors), and (b) image quality/semantic fidelity via CLIP score, FID, and related measures.
>
> We will add these instantiations to Sec. 3 so that readers can see precisely how $(Q_f, Q_0, s)$ is instantiated in practice, not just in abstract notation.
>
> **W7**. Our goal is not to argue that a unified optimization problem is always better than solving each of these problems separately. Instead, our focus is on the **definition and contract level**. In realistic deployments, a service provider may receive heterogeneous unlearning requests over time (per-user data deletion, group-level or concept-level suppression, safety-driven refusals, etc.), while still needing to communicate to users, auditors, or regulators “what kind of unlearning guarantees this system provides” in a single, coherent language. The (ε, δ, κ) framework is designed precisely for this: it provides a common vocabulary in which different request types can be specified (who/what to forget, and to what suppression level) and in which the guarantees of different mechanisms—data-removal–style or suppression-style—can be compared and certified. We will make this perspective explicit earlier in the paper by stating that we assume an environment with potentially mixed unlearning requests and that our contribution is to refine the *definition* of machine unlearning in such a setting, rather than to claim that one unified algorithm should solve all instance- and concept-level forgetting tasks. In this sense, the work on test-time privacy address yet another complementary layer of protection, while our framework clarifies how training-time unlearning guarantees should be specified and interpreted under a single behavioral contract.
>
> We have made every effort to address the reviewers’ concerns as thoroughly as possible. If there are any remaining issues or points that would benefit from further clarification, we would be very happy to address them. We are currently revising the manuscript to incorporate these changes and clarifications, and we will upload an updated version as soon as it is ready.

---

> > ### Comment · Reviewer_Xiht · 2025-11-26
> >
> > Regarding W5: Yes, more experiments with different settings are needed here.
> >
> > Regarding W6: Thanks, this indeed is helpful and strengthens the paper.
> >
> > Regarding W7: Yes, making this prominent early on is the way to go. Additionally, I'd advise to try to present a system/architecture styke of a solution. i.e. your 'umbrella system/definition' is an architecture component receiving "unlearning" requests. Your component then breaks these down into appropriate protocols, guarantees/contracts, even evaluation metrics, etc. This, I expect would help surface practical/utility aspects and position your work better(?).

---

### Official Review · Reviewer_zmtz · 2025-10-30

**Soundness:** 3
**Presentation:** 3
**Contribution:** 1
**Rating:** 2
**Confidence:** 5

**Summary:**

The paper introduces a new conceptual framework called Suppressive Machine Unlearning, aiming to unify two notions: 1) data removal unlearning (erasing the influence of specific training data) and 2) knowledge suppression (reducing a model’s ability to recall or utilize certain learned knowledge). The authors argue that knowledge suppression, while not removing data influence, serves a complementary role to unlearning by constraining model outputs or representations associated with undesired information. The paper formalizes the relation between data removal and suppression that connect suppressive behavior to approximate indistinguishability guarantees.

**Strengths:**

- The attempt to formally describe"knowledge suppression" is conceptually interesting and may inspire further discussion.
- The paper’s definitions are mathematically well-specified and traceable to standard differential privacy and approximate unlearning formulations, aiding reproducibility.

**Weaknesses:**

- The core claim that knowledge suppression constitutes unlearning is conceptually problematic. Classical unlearning (as required by GDPR’s "right to be forgotten") demands the removal of internal representations and traces of data influence, not merely output suppression. The proposed "suppressive unlearning" aligns more closely with alignment or behavioral control, which regulates model outputs without necessarily altering internal knowledge states. The authors should explicitly distinguish removal of influence from restriction of behavior and discuss them as separate but related notions.
- Lemma 4.2, Corollary 4.3, and Theorem 4.4 directly follow from standard DP properties (group-privacy, composition, post-processing). Even if these results were not explicitly stated in previous unlearning papers, their derivation is immediate from the definition of approximate unlearning (which itself mirrors DP). Therefore, these cannot be regarded as the main theoretical contribution.
- Theorem 4.4 merely restates the post-processing property of DP in the unlearning context: if the model is $(\epsilon, \delta)$-approximately unlearned, any suppressive post-processing cannot worsen privacy guarantees. This is a direct corollary, not a novel insight.
- The paper could be strengthened by analyzing relationships rather than unification. E.g., comparing unlearning (data removal) and suppression (behavior modification) as complementary processes with different guarantees.

**Questions:**

See weaknesses

---

> ### Author Response · Authors · 2025-11-24
> **[1/1] Rebuttal**
>
> We sincerely thank the reviewers for their careful reading of our paper and for the constructive feedback. Below, we respond to the comments point by point and explain how we plan to revise the manuscript accordingly.
>
> **W1**. We appreciate the distinction reviewer draw between classical “right-to-be-forgotten” style unlearning (aiming to remove internal influence of the data) and purely behavioral output control. Our goal is not to collapse these into a single notion, but to provide a framework that can represent both within a common, behavior-level contract. Classical unlearning is typically formalized by comparing the unlearned model to a retrained model on $D \setminus D_f$, and is interpreted as removing the influence of $D_f$ from internal representations. However, as models and unlearning targets have evolved (LLMs, concepts, user groups, safety-related behaviors), a growing body of work already treats “concept/knowledge suppression” as a form of machine unlearning, even when it does not operate on all parameters or fully emulate a retrain (e.g., LoRA-based unlearning that only modify a small subspace but are still framed as unlearning). These methods are not naturally captured by the classical definition, because their target is specified directly at the behavioral level, rather than via proximity to a retrained model. Our (ε, δ, κ)-suppressive definition is intended as an extension that can *explicitly distinguish* and relate these modes under a single umbrella. Our theorems then describe how these two regimes trade off and when suppression guarantees are compatible with (or implied by) classical approximate unlearning guarantees. As we discuss in Sec. 5, in many realistic deployments exact removal of all internal traces of $D_f$ is practically unattainable, whereas what users, auditors, and regulators can actually observe and verify is the model’s behavior. In this sense, certified suppression on $Q_f$ is a pragmatic form of unlearning: it operationalizes the right-to-be-forgotten at the level of observable predictions when full retraining or precise influence accounting is infeasible, while remaining aligned with the classical notion.
>
> **W2, W3**. We acknowledge that Lemma 4.2, Corollary 4.3, and Theorem 4.4 are technically derived using standard tools from differential privacy. The novelty we aim for is not in inventing entirely new mathematical techniques, but in making these relationships explicit in the machine unlearning: our results connect (ε, δ), the size of the forget set, and the behavioral suppression level κ in a way that can be directly interpreted for unlearning. In particular, Theorem 4.4 can be viewed as an adaptation of the DP post-processing property to unlearning mechanisms: if an algorithm is (ε, δ)-approximately unlearning, then any additional post-processing that only reduces utility on $Q_f$ cannot worsen this guarantee. Our contribution here is to express this implication explicitly in terms of κ and the pair $(Q_f, Q_0)$, so that one can see when a requested suppression level is compatible with a given approximate-unlearning guarantee and how changes in k or (ε, δ) inevitably constrain the behavioral suppression.
>
> **W4**. Reviewer's comments resonate with one of our main motivations. In current practice, a nontrivial number of knowledge-suppression/editing methods are explicitly framed as “machine unlearning” in top-tier venues, even though they primarily impose behavioral constraints on specific concepts or user groups rather than strict proximity to a retrained model. This suggests that, in the community, classical data-removal unlearning and suppression-based approaches are already coexisting under the same umbrella, but with somewhat ambiguous boundaries. Within our definition, these appear as two complementary regimes: classical approximate unlearning specifies guarantees relative to a retrained model and is often interpreted as internal influence removal, whereas suppression-oriented methods are captured by directly specifying a target suppression level on $Q_f$. The (ε, δ, κ) notion is intended to make their relationship explicit – e.g., when classical guarantees imply a certain level of suppression, and where pure suppression may not suffice to match retraining-like guarantees.
>
> We have made every effort to address the reviewers’ concerns as thoroughly as possible. If there are any remaining issues or points that would benefit from further clarification, we would be very happy to address them. We are currently revising the manuscript to incorporate these changes and clarifications, and we will upload an updated version as soon as it is ready.

---

> > ### Comment · Reviewer_zmtz · 2025-11-26
> >
> > Thank you for the detailed response. However, I must respectfully re-emphasize that I fundamentally disagree with framing suppression as unlearning. Even though several recent top tier papers use this terminology, I believe this trend is conceptually misleading because suppression only changes behavior and does not remove the internal influence of the forgotten data, which is central to classical and GDPR-motivated unlearning. The proposed framework is useful for analyzing behavioral guarantees, but it does not change my evaluation, and I will keep my original score.

---

### Official Review · Reviewer_thVq · 2025-11-01

**Soundness:** 3
**Presentation:** 2
**Contribution:** 2
**Rating:** 4
**Confidence:** 3

**Summary:**

This work proposed new and unified definition of machine unlearning, covering both data removal and knowledge suppression scenarios. Moreover, this work provided the bound of knowledge suppression level with data removal parameters, and the experiments are aligned with the bounds.

**Strengths:**

1. The paper proposed a new definition for machine unlearning, trying to satisfy the different types of unlearning requests.

2. The derived bound at eq 12 provides a guidance the tradeoff between utility preservation and forgetness capability.

**Weaknesses:**

1. In the discussion section, the authors discuss the challenges for LLMs; however, even for the classification task, it is impractical to retrain the models, leading the proposed definition not applicable in most of scenarios.

2.The experiments are limited, only one dataset and one model architecture are explored; moreover, how about class-unlearning scenario?

**Questions:**

1. In practice, how did the users (Alice) specifiy the kappa suppression level? As Alice won't have the access of Q_0 to estimate proper kappa?

2. How could the proposed theorems help the existed unlearning quickly find the results when kappa_u is asked? E.g., a way to guide the unlearning method instead of trial and error?
3. As BU an outlier, could the authors elaborate more what properties in the machine unlearning algorithms could result in better knowledge suppression?

---

> ### Author Response · Authors · 2025-11-24
> **[1/2] Rebuttal**
>
> We sincerely thank the reviewers for their careful reading of our paper and for the constructive feedback. Below, we respond to the comments point by point.
>
> **W1**. We agree that fully retraining modern models, especially LLMs, is often infeasible in practice. In the discussion section, our intention was precisely to highlight the *methodological* limitation, not to suggest that our definition itself is only meaningful when full retraining is operationally available. Our (ε, δ, κ)-suppressive unlearning definition is meant as a behavioral contract: it specifies how the *post-unlearning model* should behave relative to the user’s unlearning request, including an explicit suppression level κ on the forget target. It allows auditing and certification of unlearning guarantees independently of whether retraining is computationally feasible at deployment scale. This perspective is consistent with how many recent machine unlearning works are already evaluated: either by approximating the performance of a hypothetical “retrained without the forget set” baseline [1], or by declaring success when specific knowledge or concepts are effectively suppressed while retaining utility on non-forget data [2]. Our framework is designed to formalize these practices in a single, behavior-level definition that covers both data-removal–style methods and suppression-based approaches. To avoid confusion, we will clarify in the revised version that our discussion of LLMs concerns current experimental feasibility, and our definition should be read as a general behavioral and auditing framework, not as a requirement to perform full retraining in every practical scenario.
>
> [1] Yao, J., Chien, E., Du, M., Niu, X., Wang, T., Cheng, Z., & Yue, X. (2024). Machine unlearning of pre-trained large language models. *arXiv preprint arXiv:2402.15159*.
>
> [2] Park, Y. H., Yun, S., Kim, J. H., Kim, J., Jang, G., Jeong, Y., ... & Lee, G. (2024). Direct unlearning optimization for robust and safe text-to-image models. *Advances in Neural Information Processing Systems*, *37*, 80244-80267.
>
> **W2**. Our experiments focus on a class-specific setting. As described in Sec. 4.3, the forget set $D_f$ is constructed by taking k ∈ {50, 100, 500, 1000, 2000} training samples from class 0, and the forget-target distribution $Q_f$ is the class-0-only test set, while $Q_0$ is the test set excluding class 0. In other words, we study how unlearning behaves when we remove a semantically coherent subset (all from the same class) and increase the number of instances from that class. We see this as representative of many practical unlearning requests, where the forget data is concentrated on a particular semantic slice (e.g., a concept, user group, or label) rather than being uniformly scattered. Our definition is agnostic to which classes are involved.

---

> ### Author Response · Authors · 2025-11-24
> **[2/2] Rebuttal**
>
> **Q1**. κ in our framework plays a role similar to (ε, δ) in differential privacy: it is a design parameter of the unlearning mechanism, rather than something that the user must compute from $Q_0$. In a typical deployment, the model owner already knows (and often reports) the baseline utility on the retain distribution $Q_0$. Based on this baseline, one can decide how much utility on the forget target should be allowed to remain and encode this as a desired suppression level κ. In practice, a user’s unlearning request can be translated into a target suppression requirement (for instance, accuracy on Q_f close to random guessing), and the service provider then selects or configures an unlearning algorithm whose certified (ε, δ, κ) guarantee meets that requirement. This is analogous to DP, where mechanisms are chosen or tuned to satisfy a given privacy budget, without requiring the user to directly estimate ε.
>
> **Q2**. Our theorems are primarily intended as an auditing and interpretive tool rather than as a complete replacement for empirical tuning. They relate the suppression level $κ_u$ to approximate-unlearning parameters such as (ε, δ), the size of the forget set $k$, and the utility gap. For unlearning mechanisms that provide certified (ε, δ) guarantees (e.g., DP-style or certified unlearning methods), a requested $κ_u$ immediately induces constraints on the admissible (ε, δ) through our bounds. This can guide the choice of mechanism or budget, rather than relying on unconstrained trial-and-error. For deep neural models, it is indeed difficult to compute exact (ε, δ), which is why our experiments adopt the AUC-based surrogate in Sec. 4.3. But our theorems still act as a lens for interpreting the behavior of existing approximate unlearning methods: if a method effectively behaves like an (ε, δ)-unlearning mechanism (as reflected in the observed AUC gap on $Q_0$ vs. $Q_f$), our theory explains how much suppression $κ_u$ one should expect and when further suppression would necessarily trade off utility elsewhere.
>
> **Q3**. We agree that BU behaves differently from the other baselines, and this is consistent with how BU is designed. BU explicitly works in the **decision space** and aims to unlearn an entire class by *shifting the decision boundary* of the forgetting class. As a result, the logits around that semantic region are aggressively flattened regardless of the number of forget samples, which naturally leads to much stronger suppression on $Q_f$ in our definition.
>
> More broadly, BU is a representative example of **suppression-oriented unlearning** algorithms: instead of constraining parameter distance to a retrained model, they optimize a behavioral objective defined directly on a specific semantic region (e.g., a class or concept), asking the model to “not be predictive” there while preserving utility elsewhere. Recent LLM unlearning works follow the same pattern – for instance, knowledge/behavior-level unlearning that removes particular concepts or unsafe behaviors while keeping general capabilities intact. Our (ε, δ, κ)-suppressive notion is designed exactly to capture this family of methods: algorithms like BU that decisively reduce utility on $Q_f$ while maintaining acceptable performance on $Q_0$ correspond to satisfying stricter suppression requirements, which explains why BU stands out in the trade-off curves.
>
> We have made every effort to address the reviewers’ concerns as thoroughly as possible. If there are any remaining issues or points that would benefit from further clarification, we would be very happy to address them. We are currently revising the manuscript to incorporate these changes and clarifications, and we will upload an updated version as soon as it is ready.

---

### Note · Authors · 2026-01-18

**Comment:**

We sincerely thank the reviewers for their careful reading and insightful comments. Although we have decided to withdraw this submission at this stage, we greatly appreciate the feedback provided, which will be valuable in improving our work for future submission.

**Withdrawal Confirmation:**

I have read and agree with the venue's withdrawal policy on behalf of myself and my co-authors.